# Pulmonary Complications in Hematopoietic Stem Cell Transplant Recipients—A Clinician Primer

**DOI:** 10.3390/jcm10153227

**Published:** 2021-07-22

**Authors:** Anna Astashchanka, Joseph Ryan, Erica Lin, Brandon Nokes, Catriona Jamieson, Seth Kligerman, Atul Malhotra, Jess Mandel, Jisha Joshua

**Affiliations:** 1Division of Pulmonary, Critical Care, Sleep Medicine, and Physiology, University of California San Diego, La Jolla, CA 92121, USA; aastashchanka@health.ucsd.edu (A.A.); e4lin@health.ucsd.edu (E.L.); bnokes@health.ucsd.edu (B.N.); amalhotra@health.ucsd.edu (A.M.); jmandel@health.ucsd.edu (J.M.); 2Division of Hematology & Oncology, Scripps Clinic, La Jolla, CA 92037, USA; ryan.joseph@scrippshealth.org; 3Sanford Stem Cell Clinical Center, Moores Cancer Center, Department of Medicine, Division of Regenerative Medicine, University of California San Diego, La Jolla, CA 92093, USA; cjamieson@health.ucsd.edu; 4Division of Cardiothoracic Radiology, University of California San Diego, La Jolla, CA 92121, USA; skligerman@health.ucsd.edu

**Keywords:** post-HSCT, PERDS, DAH, IPS, bronchiolitis obliterans syndrome

## Abstract

Hematopoietic stem cell transplants (HSCT) are becoming more widespread as a result of optimization of conditioning regimens and prevention of short-term complications with prophylactic antibiotics and antifungals. However, pulmonary complications post-HSCT remain a leading cause of morbidity and mortality and are a challenge to clinicians in both diagnosis and treatment. This comprehensive review provides a primer for non-pulmonary healthcare providers, synthesizing the current evidence behind common infectious and non-infectious post-transplant pulmonary complications based on time (peri-engraftment, early post-transplantation, and late post-transplantation). Utilizing the combination of timing of presentation, clinical symptoms, histopathology, and radiographic findings should increase rates of early diagnosis, treatment, and prognostication of these severe illness states.

## 1. Introduction

Hematopoietic stem cell transplant (HSCT) involves replacing a patient’s bone marrow with hematopoietic stem or progenitor cells from peripheral blood, bone marrow, or umbilical cord from another person or the same individual to restore immune–hematopoietic function after the underlying disease is eliminated [1]. Pulmonary complications post-HSCT affect between 45% and 60% of recipients [2,3] with a mortality rate exceeding 60% in mechanically ventilated patients after autologous HSCT [4]. This review is intended to form a framework for diagnosing and treating non-infectious and infectious pulmonary complications post-HSCT for the general clinician.

## 2. Hematopoietic Stem Cell Transplant Overview

Hematopoietic stem cell transplant (HSCT) is a multi-step process with a high risk for complications during marrow ablation, during engraftment, or afterwards [5,6,7]. Successful transplantation depends on the selection of the hematopoietic stem cell source, host preparation (conditioning regimen), and modulation of immune cell engraftment to minimize graft-versus-host disease (GVHD) [7].

### 2.1. Autologous versus Allogeneic Hematopoietic Stem Cell Transplantation (HSCT)

Hematopoietic stem cells can be obtained from self (autologous or auto-HSCT) or from a donor (allogeneic or allo-HSCT). Deciding whether to use autologous versus allogeneic transplant is dependent on multiple patient-specific factors and donor availability [8]. Autologous HSCTs have many advantages: no need to identify human leukocyte antigen (HLA)-matched donor, no risk of GVHD, no need for immunosuppressive therapy, more rapid hematopoietic recovery post-HSCT, and a lower risk of opportunistic infections. However, the relapse rate of underlying malignancy after autologous HSCT is often higher than in allogeneic HSCT, as autologous HSCT relies on the tumor response from cytoreductive therapy alone and does not induce a graft versus tumor response. 

Advantages of allogeneic transplant include the ability to correct congenital or acquired defects and the immunologic, anti-tumor effects that can occur in response to persistent disease referred to as the graft versus tumor (GVT) effect [9]. Allogeneic transplant requires that the products be HLA-matched, as poorly HLA-matched products can lead to immune dysregulation and increased risk of GVHD. Furthermore, immune recovery is slower, and opportunistic infections are more common.

### 2.2. Engraftment

After selecting the source of stem cells, the host is prepared via high dose myeloablation therapy, reduced intensity myeloablation, or non-myeloablative conditioning [10]. Once a patient undergoes HSCT, engraftment occurs within a conditioning and host-dependent timeframe. 

While there are several definitions of successful engraftment, patients typically develop a neutrophil count greater than 500/mm^3^, a platelet count greater than 20,000/microliter without any transfusions for one week, and a hematocrit greater than 25% for at least 20 days without any transfusions [11]. For autologous HSCT, white blood cell recovery typically occurs within two weeks while red blood cell and platelet recovery varies from patient to patient [6]. For allogeneic HSCT, peripheral blood granulocyte counts usually show signs of recovery within three weeks while platelet recovery is often delayed, taking on average 5–7 weeks [6]. Neutrophil engraftment is also dependent on the immunosuppressive regimen used, and recovery can average 10–30 days [12].

## 3. Timeline of Complications Following HSCT

The diagnosis and management of post-HSCT pulmonary manifestations requires a multidisciplinary approach. Diagnosis can be challenging as post-HSCT syndromes often have nonspecific clinical presentations. Utilizing common timelines of illness presentation post-HSCT (Figure 1) in combination with exposures to infectious agents, prior anti-microbial use, and unique radiographic findings aid in the diagnosis. Use of non-invasive tests including upper respiratory cultures, respiratory pathogen PCR, and serologies are useful adjuncts and can take the place of invasive testing to identify infectious pathogens [13]. Timely coordination and open communication between primary service, hematology–oncology, pulmonary, radiology, pathology, and infectious disease specialists is vital to the management of these patients. 

Infectious and non-infectious pulmonary complications are classified by etiology and temporality in reference to HSCT, as it reflects the immunologic state of the patient. The often-cited time course for non-infectious entities consists of the pre/peri-engraftment phase (first 30 days), early post-transplantation phase (30–100 days), and the late post-transplantation phase (after 100 days). Infectious complications will be discussed separately.

## 4. Peri-Engraftment Period (0–30 Days Post-Transplant)

Conditions in the peri-engraftment period occur secondary to neutropenia, transfusion-related events, and toxicity of conditioning regimens. Non-infectious and infectious etiologies are equally likely to develop and often co-occur. An attempt should be made to identify and treat infection prior to initiation of immunosuppressive treatment for non-infectious complications; however, in critically ill patients, infectious and non-infectious complications may require concurrent treatment. 

Cardiogenic causes of acute lung injury include volume overload resulting from large fluid volumes administered during chemotherapy, cardiac toxicity due to prior chemotherapy or radiation therapy, and transfusion-associated circulatory overload (TACO) after multiple blood products are transfused. Non-cardiogenic causes include chemotherapy pneumonitis and transfusion-related acute lung injury (TRALI). Chemotherapy-associated pneumonitis can occur due to a number of different chemotherapeutic agents including melphalan, busulfan, and bis(chloroethyl)nitrosourea (BCNU) [14]. Drug associated pneumonitis is often treated conservatively with withdrawal of offending agents and/or high-dose corticosteroids. TRALI should also be considered in patients requiring frequent blood product transfusions while awaiting engraftment. TRALI is typically seen within 6 h post-transfusion with acute respiratory failure, often requiring mechanical ventilation. Risk of TRALI is mitigated by conservative transfusion thresholds and protocols aimed at using low alloantibody products [15].

### 4.1. Peri-Engraftment Respiratory Distress Syndrome (PERDS) 

Peri-engraftment respiratory distress syndrome (PERDS) has a wide range of reported incidence with a propensity to occur more frequently in allogeneic HSCT [9]. It is one of the earliest non-infectious pulmonary complication, occurring on average 7 days post-transplant with an average time to onset between 5 and 21 days [9]. The etiology is related to release of pro-inflammatory cytokines and early neutrophil engraftment, manifesting within 96 h of neutrophil recovery. PERDS has also been reported as an early sign of graft rejection, especially in cyclophosphamide-based non-myeloablative HSCTs with PERDS occurring independently from graft-versus-host disease (GVHD) [9]. PERDS is the pulmonary manifestation of engraftment syndrome and may be associated with multi-organ dysfunction. The Spitzer diagnostic criteria for engraftment syndrome include major criteria of non-infectious fever, erythrodermatous rash, and non-cardiogenic pulmonary edema, with minor criteria of weight gain, hepatic/renal dysfunction or transient encephalopathy [9]. Imaging during the early phase of PERDS demonstrates diffuse ground-glass opacities often with septal thickening and small pleural effusions (Figure 2a). 

While most patients with PERDS respond favorably to treatment, some demonstrate progressive acute lung injury with lung consolidation and increasing alveolar collapse requiring intubation (Figure 2b,c). Up to one-third of patients also develop diffuse alveolar hemorrhage (DAH) [16]. Patients with respiratory failure requiring mechanical ventilation have a worse prognosis. Treatment consists of a short course of corticosteroids, with reports of rapid improvement within 24–48 h of corticosteroid initiation [9,16]. 

### 4.2. Diffuse Alveolar Hemorrhage (DAH)

Diffuse alveolar hemorrhage (DAH) has similar rates in both allogeneic (3–7%) and autologous (1–5%) HSCT recipients [17,18,19,20,21,22,23,24,25,26,27,28]. While classically considered an early pulmonary complication occurring within a month of transplant, up to 30% of cases of DAH occur >100 days post-transplantation [20]. The pathogenesis of DAH remains unclear and may be secondary to disruption of the alveolar–capillary basement membrane by conditioning, immune-mediated events, and return of neutrophils during marrow recovery. Older age, cord blood as transplant source, allogeneic HSCT, myeloablative conditioning, and acute GVHD are all considered risk factors for DAH [20,21].

DAH presents with fever, dyspnea, non-productive cough, and diffuse pulmonary infiltrates. Hemoptysis is relatively rare, making it a non-reliable sign for diagnosis. Computed tomography (CT) of the chest usually demonstrates patchy multifocal or diffuse ground-glass opacity with superimposed interlobular and intralobular septal thickening, creating a “crazy paving” pattern (Figure 2d). Bronchoscopy is diagnostic of DAH and is characterized by bloodier returns during bronchoalveolar lavage (BAL) with each instilled aliquot from at least three different samples from separate segmental bronchi or ≥20% hemosiderin-laden macrophages in the absence of infectious cause [18]. The majority of these patients require intensive care unit (ICU) admission and mechanical ventilation [18]. Treatment consists of high-dose corticosteroids. Despite treatment, patients with DAH have a poor prognosis with up to 80% case fatality [19,21]. Prognosis is better in those who present early after HSCT and in those receiving auto-HSCT (<30% mortality) compared to those observed later or in an allo-HSCT setting (>75% mortality) [18]. Additionally, underlying coagulopathy, need for mechanical ventilation, and identification of an organism on BAL are also associated with higher mortality and overall poor outcome [19,21]. 

## 5. Early Post-Transplantation Period (31–100 Days Post-Transplant)

This period is characterized by impaired cellular and humoral immunity and delayed lung injury from conditioning/preparative regimens. During this phase, the neutrophil count is generally normal, and the frequency of infectious complications typically decreases. Delayed complications of radiation and chemotherapy can produce manifestations similar to other HSCT-related complications during this period. 

### 5.1. Idiopathic Pneumonia Syndrome (IPS)

Idiopathic pneumonia syndrome (IPS) represents a spectrum of diseases occurring between 20 and 42 days post-transplant, with the range extending up to 100+ days post-transplant. It is more common in allogeneic HSCT recipients [22]. Myeloablative conditioning with high-dose total body irradiation and acute GVHD are major risk factors for IPS development [24,25,26,27]. Patients often present with fever, non-productive cough, tachypnea, and hypoxemia. The diagnosis is based on three major requirements: widespread alveolar injury with symptoms and signs of pneumonia; absence of active lower respiratory tract infection; and absence of cardiac dysfunction, acute renal failure, or iatrogenic fluid overload as an etiology [26]. Imaging is non-specific, presenting as ground-glass opacity or consolidation in severe courses [27], which is often indistinguishable from other causes of acute lung injury (Figure 3a). 

While both PERDS and DAH technically meet the diagnostic criteria of IPS, the three disease entities behave differently. High-dose corticosteroids continue to be the treatment of choice with variable response. Up to 85% of IPS cases are non-responsive to corticosteroids and progress to respiratory failure with up to 90% requiring mechanical ventilation [22,28]. Overall, IPS has a very poor prognosis with 60–80% case fatality [22,28,29,30]. The addition of etanercept has demonstrated promising early results with an overall improvement of 2-year survival although follow-up studies have suggested less durable benefit [30,31,32]. 

### 5.2. Pulmonary Cytolytic Thrombi

Pulmonary cytolytic thrombi are intravascular thromboemboli accompanied by the infiltration of monocytes into small and medium pulmonary vessels surrounded by areas of hemorrhage or infarct [33]. This phenomenon is most often seen post-allogeneic HSCT, primarily in children. It occurs a median 80 days, but can occur up to a year after transplant [34]. It is associated with acute and chronic GVHD. Symptoms are non-specific including fever, dyspnea, and chest pain. CT findings show multiple bilateral pulmonary nodules and peripheral opacities. Diagnosis of pulmonary cytolytic thrombi is made via open lung biopsy and cannot be treated empirically [34]. Treatment involves increasing immunosuppression, and the condition usually resolves within weeks to months [33].

### 5.3. Post-Transplant Lymphoproliferative Disease (PTLD)

Post-transplant lymphoproliferative disease (PTLD) is a heterogenous group of lymphoid disorders that can complicate solid organ transplants as well as HSCTs. While in solid organ transplants, most PTLD is host-derived, the majority of PTLD (55–65%) in HSCT is due to reactivation and proliferation of Epstein–Barr virus (EBV)–infected donor derived B-lymphocytes due to weakened cytotoxic T-cell immunity [35]. However, EBV-negative cases of PTLD have been reported irrespective of donor or recipient’s EBV-status [36]. These can be differentiated into multiple subgroups: non-destructive PTLDs (including plasmocytic hyperplasia, infectious mononucleosis, and florid follicular hyperplasia), polymorphic PTLD, monomorphic PTLDs, and classical Hodgkin lymphoma PTLD [37]. It presents within 100 days in 75% of patients [38]. Risk factors for EBV-positive PTLD are directly related to the degree of T-cell depletion or impairment and thus include unrelated or human leukocyte antigen (HLA)-mismatched donors [39], EBV seropositivity status mismatch [39], cytomegalovirus (CMV) reactivation as a variable of EBV reactivation [40], splenectomy, and duration of immunosuppression [41]. Imaging often shows multifocal lymphadenopathy (Figure 3b,c). The leading treatment strategy is reduction of immunosuppression, with benefit seen with rituximab and chemotherapy in non-responders [35]. Survival is poor with an 80% mortality rate at 3 years [41].

## 6. Late Post-Transplantation Period (>100 Days Post-Transplant)

The term late-onset non-infectious pulmonary complications (LONIPCS) has been used to describe complications that occur 3 months after transplant [42]. These are caused by delayed immune recovery, immunosuppression, and immune-mediated reaction damage to the host tissue by the donor cells. Guidelines by the American Society of Blood and Bone Marrow Transplantation recommend pulmonary function testing (PFTs) at 6 months and then yearly post-HSCT for early diagnosis and treatment of LONIPCS including most notably bronchiolitis obliterans syndrome (BOS) [43].

### 6.1. Bronchiolitis Obliterans Syndrome (BOS)

Chronic graft-versus-host disease (cGVHD) is currently the leading cause of long-term morbidity and mortality post-allogeneic HSCT [44]. Clinical diagnosis of bronchiolitis obliterans syndrome (BOS), in addition to one other distinctive manifestation of cGVHD, is sufficient to establish the diagnosis of cGVHD [45]. 

BOS is a clinical rather than a histopathological diagnosis [46]. BOS is characterized by (1) fixed airflow obstruction with forced expiratory volume in 1 s (FEV1)/vital capacity (VC) ratio <0.7 or the 5th percentile of predicted, (2) FEV1 < 75% of predicted with ≥10% decline in less than 2 years, (3) in the absence of respiratory tract infection (4) with signs of air trapping on PFTs or signs of air trapping or small airway thickening or bronchiectasis on CT chest [32]. BOS occurs between 6–24 months after transplant and occurs almost exclusively in allogeneic HSCT recipients [47]. The most common symptoms are progressive dyspnea, non-productive cough, and wheezing [46], although onset is generally insidious with up to 20% of patients remaining asymptomatic at diagnosis [47]. Imaging demonstrates a mosaic pattern on the inspiratory scan (Figure 4a,b). On expiratory imaging, the abnormal areas with air-trapping will not change in attenuation while normal lung will increase in attenuation (Figure 4c). The role of bronchoscopy is limited, often resulting in non-diagnostic findings of neutrophilic and/or lymphocytic inflammation [47]. As transbronchial biopsy has low sensitivity, the gold standard is open lung biopsy, although it is rarely performed. When performed, BO is characterized by submucosal bronchiolar fibrosis and luminal narrowing affecting the small airways. 

BOS is irreversible with variable progression and mortality rates of up to 60% [46]. Management remains a challenge. Budesonide/formoterol, when compared to placebo, improved FEV1 by an average of 260 mL over one month although patients did not experience significant improvement in their respiratory symptoms [48]. Fluticasone, azithromycin, and montelukast (FAM regimen) with a brief steroid pulse suggested stabilization of pulmonary function tests [49] and is currently recommended as initial treatment of BOS [50]. The utility of azithromycin is unclear with a study showing no improvement in respiratory function with its use [51]. Furthermore, early azithromycin administration in patients undergoing allogeneic HSCT is associated with worse airflow decline–free survival [48]. A recent case series found that ruxolitinib in patients with steroid refractory cGVHD had an overall response rate of 70.7%; however, the response is less favorable in patients with lung involvement [52]. A randomized clinical trial of ruxolitinib in treatment of cGVHD is currently underway (NCT03112603). Extracorporeal photodynamic therapy may be associated with better overall survival [53]. Lung transplantation may also be an option for eligible candidates if medical therapy is unsuccessful [47].

### 6.2. Organizing Pneumonia

Organizing pneumonia (OP) has an incidence of approximately 10% after HSCT [54]. It occurs in both autologous and allogeneic HSCT, but it is more common in allogeneic recipients and can occur as a manifestation of cGVHD [55,56]. Symptoms include fever, shortness of breath, and non-productive cough with a median onset of 108 days [56]. Histopathologic confirmation is the gold standard for diagnosis with pathology demonstrating plugs of granulation tissue and fibrosis in the distal airways and alveoli with accompanying interstitial and alveolar inflammation [57]. There are numerous patterns of OP on CT, but common appearances include subpleural and peribronchovascular areas of ground-glass opacity, consolidation and linear opacities [58]. The “reverse halo” or “atoll” sign, due to areas of curvilinear consolidation with adjacent ground-glass opacity may also be present (Figure 4c) [58]. OP is very responsive to corticosteroid therapy [57], but recurrence is not uncommon [56]. Its prognosis is more favorable than BOS [57].

### 6.3. Interstitial Lung Disease

More recently, studies have focused on the prevalence of interstitial lung diseases (ILD) after HSCT [59]. While the incidence of non-infectious ILD after allo-HSCT is as low as 2–3% [60,61], ILD contributes to approximately 20% of LONIPCS [61]. Patients present with non-specific symptoms of shortness of breath and cough [61]. The main pulmonary function test (PFT) abnormality is a restrictive ventilatory defect [61]. Radiographic and pathologic findings vary, depending on the underlying entity. A histopathologic study including specimens from 18 different sites showed a variety of patterns of their interstitial lesions including non-specific interstitial pneumonia (NSIP), lymphoid interstitial pneumonia (LIP), diffuse alveolar damage (DAD), and pleuroparenchymal fibroelastosis (PPFE) [62].

One example of post-HSCT ILD is pleuroparenchymal fibroelastosis (PPFE), a rare form of interstitial pneumonia. PPFE can be idiopathic or secondary, the latter of which has been associated with HSCT. It occurs in <0.5% of patients post-HSCT and typically presents many years post-transplantation [63,64,65]. The incidence is higher after allogeneic HSCT [66]. This may occur secondary to diffuse alveolar damage with a failure of parenchymal lung injury to adequately resolve, promoting aberrant tissue repair [63]. CT imaging shows prominent superior pleural thickening often extending inferiorly to involve the major fissures. There is associated upper lobe fibrosis with superior displacement of the hila and traction bronchiectasis (Figure 4d–f). A histopathologic diagnosis of PPFE requires demonstration of interalveolar fibrosis and elastosis. In a review of 16 studies, Higo et al. also noted a high coexistence of pathologic findings of BO in addition to PPFE after allo-HSCT [66]. Pulmonary function tests typically show a restrictive ventilatory defect [65]. Several pharmacologic treatments including corticosteroids, everolimus, pirfenidone, and imatinib have been trialed with minimal success [67]. The most definitive treatment remains lung transplantation [67]. Overall, the prognosis is poor [65]. 

### 6.4. Pulmonary Veno-Occlusive Disease (PVOD)

Pulmonary veno-occlusive disease (PVOD) is a rare variant of pulmonary arterial hypertension (PAH) in which intimal proliferation occurs in the small pulmonary venules. A recent autopsy series suggests that it may be under-recognized with pathologic evidence of PVOD in 12 of the 35 autopsies in allo-HSCT recipients [68]. The etiology of PVOD has not been fully elucidated, but endothelial injury from cytotoxic chemotherapy or irradiation may be a risk factor [69]. It occurs in both autologous and allogeneic transplant recipients with time to onset ranging from several weeks to months [70]. PVOD is characterized by PAH with normal pulmonary capillary wedge pressure and pulmonary edema in the absence of left sided heart failure [69,70]. Histopathologic diagnosis may be required in patients who have an acceptable risk for biopsy [70]. Pathology is characterized by intimal proliferation and fibrosis of pulmonary venules and small veins, leading to progressive vascular obstruction and increased pulmonary capillary and arterial pressures [70]. Imaging demonstrates interstitial lung edema, diffuse ground-glass opacities with a centrilobular distribution, adenopathy, and prominent septal lines [71]. PFTs often show a severely low diffusing capacity for carbon monoxide (DLCO). Treatment remains controversial, given the risk for worsening pulmonary edema with the use of advanced pulmonary hypertension therapy. The only durable treatment is lung transplantation [70]. 

## 7. Infectious Complications

The peri-engraftment phase is hallmarked by mucositis due to conditioning therapy and prolonged neutropenia due to non-functional transplanted marrow [72]. The risk of infection is highest when absolute neutrophil count (ANC) is less than 500 and increases with increasing duration of neutropenia before engraftment [73]. During the early post-transplantation phase, infections are related to the defects in cellular immunity caused by immunosuppressive and conditioning regimens [72]. In the late post-transplantation phase, infection rates decline as immunosuppressant medications are tapered unless needed for cGVHD. Because of differences in immune recovery, autologous HSCT recipients are at higher risk of infectious complications during the peri-engraftment and early post-transplantation phase while allogeneic HSCT recipients are at higher risk of infection throughout the late post-transplantation phase due to cGVHD and prolonged immunosuppressive therapy [8,72].

### 7.1. Bacterial Pneumonia

Bacterial infections account for approximately 90% of infections during the peri-engraftment period after allo-HSCT [74] and include bloodstream, respiratory, and gastrointestinal infections [75]. The sources of most bacterial infections in allo-HSCT recipients during peri-engraftment includes the gastrointestinal tract and indwelling vascular catheters [76]. Most studies report a high incidence of bacterial pneumonias [75,77,78] with up to 15% in in allogeneic and autologous HSCT recipients [79]. A nationwide study of allo-HSCT recipients revealed that 44% of pneumonias were due to bacterial etiology, compared to 29% of fungal and 19% of viral pneumonias [80]. Lung infections that occur in late post-transplantation phase are seen in patients with cGVHD and are frequently due to encapsulated organisms, particularly Streptococcus pneumoniae and Haemophilus influenzae (Figure 5b). Rarely, tubercular and non-tubercular mycobacterial infections can be seen in the post-transplant period especially in patients with poor immune reconstitution and cGVHD (Figure 5c,d) [81].

### 7.2. Fungal Pneumonia

Invasive fungal infections (IFI) can affect patients early in their post-transplant course. Invasive aspergillosis is the most common invasive fungal infection HSCT recipients [82,83]. The incidence is higher in allogeneic than in autologous HSCT recipients [83]. There are three separate “at risk” periods, with first peak around peri-engraftment associated with neutropenia, second peak between 40 and 70 days associated with acute GVHD, and late peak late after transplant with cGVHD [84]. Patients may present with fever, cough, and hemoptysis [85]. The classic radiographic finding on non-contrast CT of the chest are nodules surrounded by ground-glass opacity (“halo sign”), representing angioinvasion and hemorrhage in the surrounding tissue (Figure 5a) [83]. The addition of contrast to the CT can be useful for detection of the angioinvasive growth pattern of *Aspergillus* and even *Mucor* and generally includes a vessel occlusion sign, which was noted to have a sensitivity of 0.78 and specificity of 0.91 for angioinvasive disease in one study [86]. The diagnosis can be difficult and rely on clinical suspicion, radiographic presentation, laboratory studies (including serum or BAL galactomannan) and possible histopathologic findings [83]. Given its high incidence, many institutions preemptively utilize fungal prophylaxis in patients on immunosuppression therapy for cGVHD [87]. Posaconazole is the prophylactic medication of choice, as it is well tolerated and covers *Aspergillus* strains resistant to more commonly used azoles [82]. Voriconazole is the preferred primary therapy for invasive aspergillosis while liposomal Amphotericin B, isavuconazole, and posaconazole can be used as alternative therapies.

Zygomycosis is the second most common mold infection in HSCT with an increasing incidence over the past few decades [85]. Rare infections with fusarium and scedosporium can occur as well [88]. Invasive fungal infections with mucormycosis generally occurs > 3 months post-transplant and can make up to 8% of fungal infections in this patient population [89]. Although there are various imaging appearances of mucormycosis, one nearly pathognomonic imaging manifestation is the “bird’s nest” sign (Figure 5e) [90]. While mucormycosis infections are rare, they are frequently fatal and require early, aggressive management. Treatment involves combination of medical (with amphotericin B for zygomycosis) and surgical therapy [88].

Candidal infections occur less frequently as a result of universal prophylaxis with fluconazole. In high-risk patients, posaconazole may be the preferred prophylactic therapy due to the emergence of resistant non-albicans candida species including *Candida glabrata* and *Candida krusei*, especially in those with cGVHD [91]. Similarly, *Pneumocystis jiroveci pneumonia* (PJP) has a low incidence due to use of prophylaxis. Without adequate prophylaxis, it has an incidence of 5–16% [92] and often presents in late post-transplantation period if prophylaxis has been discontinued [93]. CT findings can vary but include diffuse infiltrates, consolidation in the upper lobes, and ground glass. Prophylaxis includes trimethoprim/sulfamethoxazole (TMP-SMX), pentamidine or atovaquone [92]. TMP-SMX is also the preferred treatment for PJP pneumonia [88].

### 7.3. Viral Pneumonia

Viral pneumonia from reactivation is common during the peri-engraftment phase and should be closely monitored while patients are immunosuppressed. Viral infection from herpes simplex virus (HSV)-1 and -2 occur in peri-engraftment and early post-transplantation phase, those from cytomegalovirus (CMV) and human herpes virus-6 (HHV-6) are often present in early post-transplantation phase, and infection from varicella zoster virus (VZV) is more common in the late post-transplantation phase [94]. The incidence of reactivation of viruses such as HSV, CMV, and VZV has drastically decreased since prophylaxis became routine.

CMV pneumonia has an incidence as high as 20% in allo-HSCT recipients [95]. In contrast, in a review of 795 auto-HSCT patients, only 16 (2%) were diagnosed with CMV pneumonia [96]. There is a higher frequency in CMV seropositive auto-HSCT recipients, compared to seronegative [96,97]. The diagnosis of CMV pneumonia is based on the combination of clinical presentation, radiologic evidence, and virus in BAL. CMV pneumonia presents with non-productive cough, dyspnea, and hypoxemia; fever is not always present [96]. Diagnosis of CMV pneumonia is difficult. CMV viremia in the blood may be helpful but is not diagnostic. Bronchoscopy is the most common method of confirming CMV in the lungs [98,99]. Although its presence is rare, inclusion bodies in BAL have a high sensitivity but low specificity for CMV pneumonia. Immunostaining for anti-CMV antibodies in BAL fluid has both high sensitivity and specificity and may be a helpful modality for diagnosis of CMV disease [100]. The absence of both findings can exclude CMV pneumonia with up to 99% specificity [100]. Viral shedding into the respiratory tract has been demonstrated without presence of invasive disease and should be interpreted cautiously [78]. CT findings of CMV pneumonia is nonspecific and may present with a mixture of patterns, most commonly ground-glass opacities, pulmonary nodules, and areas of consolidation [101]. The standard of therapy for CMV pneumonia is ganciclovir. The rate of mortality can be high with reports of 31–62% CMV pneumonia-related mortality [92,102] although the outcome has improved over the past few decades [103]. Lymphopenia prior to diagnosis and mechanical ventilation are associated with an increased overall mortality [72]. Given the high mortality rate, all CMV seropositive HSCT recipients and all CMV seronegative recipients with CMV seropositive donors should be placed on a CMV prevention program [89]. The prevention program is generally either universal prophylaxis with either ganciclovir, high-dose acyclovir, or valacyclovir for the first 100 days, or pre-emptive strategy of weekly monitoring and early therapy [103].

HSV can have pulmonary manifestations including tracheobronchitis and pneumonia [104]. The breadth of clinical presentation from an HSV infection is quite broad—from a simple cough to respiratory failure from diffuse alveolar damage or necrotizing pneumonia [105,106]. Hemoptysis can also occur, typically due to airway inflammation in tracheitis [106]. HSV pneumonia occurs in peri-engraftment and early post-transplantation phase. The mode of transmission is from continuous oropharyngeal spread [107]. Acyclovir prophylaxis should be offered to all HSV-seropositive allogeneic recipients to prevent HSV re-activation. This is initiated at the start of conditioning and continued until engraftment occurs or until mucositis resolves. Acyclovir prophylaxis for VZV seropositive allogeneic and autologous recipients is also routinely recommended to prevent recurrent VZV infection in the first year after HSCT [84].

Patients continue to be at high risk for seasonal community-acquired respiratory viruses (CARV). Respiratory syncytial virus (RSV), influenza A and B, parainfluenza, and rhinovirus cause the majority of non-CMV viral respiratory infections [108,109], especially in allo-HSCT recipients [90,92]. Routine immunization with inactivated influenza vaccine after HST can be helpful in preventing and minimizing infection severity [110,111]. A single-center study of 195 allo-HSCT reported a 30% incidence of severe viral pneumonia causing respiratory failure and requiring mechanical ventilation [109]. The overall mortality rate is high at 15–30%, depending on the offending viral agent [108,112].

RSV is one of the more common CARV that is associated with high mortality rate. In a single center retrospective study of 280 allo-HSCT recipients with RSV infection, 29% had progression to lower respiratory tract infection [113]. Older age, smoking history, conditioning with high-dose total body irradiation, and lymphopenia were associated with an increased risk of progression from upper respiratory to lower respiratory tract infection [113,114]. RSV is associated with a wide mortality rate, ranging from 5% to 43%, in HSCT patients [112,113,115,116,117]. Older age, male gender, bone marrow or cord blood as transplant source, corticosteroid use, lower respiratory tract infection, and oxygen requirement are all associated with increased overall mortality [113,116,117,118].

In 2019, the emergence of severe acute respiratory syndrome coronavirus 2 (SARS-CoV-2) led to a coronavirus disease 2019 (COVID-19) global pandemic. The majority of HSCT patients who contract COVID-19 have mild or moderate disease [119]. Studies have demonstrated that 14–25% of HSCT patients diagnosed with COVID-19 develop severe disease with respiratory failure requiring mechanical ventilation [119,120,121]. Patients typically present with fever, cough and shortness of breath [120]. Abnormal laboratory studies including anemia and lymphopenia may be present [120]. The most common radiographic abnormality in these patients is bilateral infiltrates [120]. A mortality rate of 16–21% has been reported in HSCT patients with COVID-19 [120,121,122,123]. Older age, male gender, use of immunosuppressive agents, and COVID-19 diagnosis within 1 year of HSCT are all associated with a higher risk of mortality in HSCT patients [119,120,122].

Updated guidelines have been published by European Society for Blood and Marrow Transplantation (EBMT) and American Society for Transplantation and Cellular Therapy (ASTCT) for the management of HSCT patients in regard to COVID-19 [124,125]. Patients should follow the same practices recommended for the general public including hand hygiene, masking, and social distancing [125]. Patients with upper or lower respiratory symptoms should undergo PCR testing for SARS-CoV-2 in addition to testing for other CARV [124]. Additional chest imaging may be required in post-HSCT recipients with lower respiratory symptoms with a negative SARS-CoV-2, given the discrepancy between upper and lower respiratory tract positivity [124]. Prolonged viral shedding may be seen in post-HSCT patients, as is seen with other viral infections in this patient population and may not equate to active infection [124]. Furthermore, HSCT patients may not mount an effective humoral response, so providers need to be cautious when interpreting serologic IgG and IgM assays [124]. Management of active COVID-19 infection for HSCT patients is similar to that of the general population [124]. There are no specific therapies recommended for HSCT-recipients with COVID-19. Prophylactic therapies should be continued in patients with active infection [124]. It is recommended that immunosuppression be carefully titrated. While no cases of graft rejection have been reported as a result of COVID-19, immunosuppression is recommended for anyone that demonstrates histologic evidence of GVHD regardless of COVID-19 infection [124,125].

## 8. Diagnostic Tools

Open communication between these primary services and these subspecialists is necessary for the initial evaluation of these patients. There are multiple diagnostic tools available if non-invasive methods do not yield a diagnosis. The most frequently utilized and widely considered safest diagnostic tool is flexible bronchoscopy. Bronchoscopic findings result in identification of infectious pathogens in 31–71% cases and change in therapy in 24–52% of patients [126,127]. Although yield is higher if performed early on and prior to initiation of anti-microbials, bronchoscopy may yield a treatment-altering diagnosis in up to half of cases, [128,129] even while patients are receiving empiric antimicrobial therapy [6]. Bronchoscopy with BAL can also identify organisms not evident on non-invasive testing and proves to be an important complementary strategy in the work-up of pulmonary complications in this population [130]. Furthermore, BAL cellularity can also help with the diagnosis of non-infectious entities including DAH. However, bronchoscopy becomes higher risk in those with hypoxemia, neutropenia and/or thrombocytopenia [131]. Most complications are reported as mild and self-limiting, but more severe adverse events like respiratory failure, arrhythmias, shock, and severe bleeding occurred at a rate of ≤5% [132].

While prior studies have shown that transbronchial biopsies (TBBx) can increase the diagnostic yield compared to bronchoscopy alone [133], more recent studies note that the addition of TBBx is not associated with a change in antibiotic therapy. However, positive results did increase the odds of a change in corticosteroids, suggesting a utility in non-infectious etiologies [134]. Not surprisingly, transbronchial biopsy confers an increased rate of complications and procedural mortality compared to bronchoscopy alone [134]. Endobronchial ultrasound guided transbronchial needle aspiration (EBUS-TBNA) has been demonstrated to be a safe and effective diagnostic method in sampling peripheral ground-glass opacities with high suspicion for primary malignancy [135,136], although the study of this technique as a diagnostic tool for infectious or non-infectious complications has been limited in immunocompromised patients [137]. However, limited data does demonstrate increased yield utilizing EBUS-TBNA compared to BAL alone in this pediatric population [137]. More research is required to elucidate the benefits and risks of EBUS-TBNA in the adult HSCT population. Trials comparing yield and safety of newer bronchoscopic interventions such as cryobiopsy to traditional TBBx in HSCT patients are lacking.

The use of other more invasive modalities is controversial [138]. Percutaneous CT-guided transthoracic lung biopsy has been performed in the past [139]. Surgical lung biopsy, either by video-assisted thoracoscopic surgery (VATS) or open thoracotomy [140,141], is now rarely performed due to risk of increased post-operative procedural complications and procedure-related mortality. The utility of these more invasive techniques has decreased over the past decade as other non-invasive diagnostic tests have improved [138]. Overall, the choice of invasive modality is individualized based on risks and benefits of the procedure for the patient [142]. The decision for and mode of bronchoscopic evaluation is institution- and practitioner-specific [142].

## 9. Summary

Collectively, the risk of both infectious and non-infectious pulmonary complications post-HSCT are high, but careful consideration of the timeline post-HSCT, an assessment of individual and local risk factors, and early involvement of a multidisciplinary team in the care of these individuals can help to identify and treat these complications in a timely manner.

## Figures and Tables

**Figure 1 jcm-10-03227-f001:**
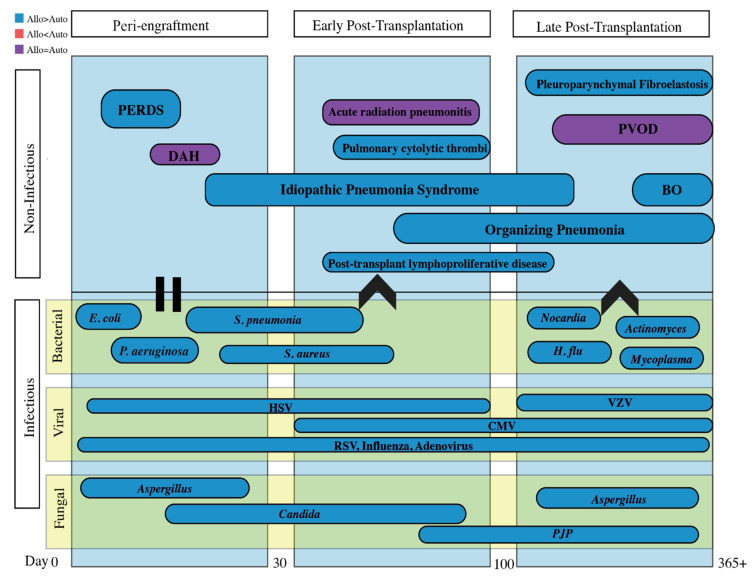
A timetable organized by peri-engraftment, early post-transplantation and late post-transplantation periods encompassing both infectious and non-infectious pulmonary complications. Important to note that all of the above vary in time of presentation and can often overlap. For example, the increase in non-myeloablative conditioning can make this timeline less reliable. Infectious and non-infectious complications tend to occur more commonly in allogeneic transplants are typically secondary to chronic graft-versus-host disease and prolonged immunosuppressive medications. Abbreviations: PERDS = peri-engraftment respiratory distress syndrome; DAH = diffuse alveolar hemorrhage; PVOD = pulmonary veno-occlusive disease; BO = bronchiolitis obliterans; HSV = herpes simplex virus; VZV = varicella zoster virus; RSV = Respiratory syncytial virus; CMV = cytomegalovirus; PJP = Pneumocystis jirovecii.

**Figure 2 jcm-10-03227-f002:**
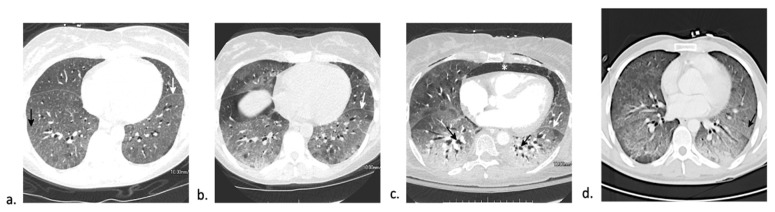
CT findings of peri-engraftment non-infectious pulmonary complications. a–c: Peri-engraftment respiratory distress syndrome (PERDS) in a 28-year-old woman with a fever and rash who is 7 days post-autologous HSCT for acute myelogenous leukemia (AML). (**a**) Axial image from a chest CT shows diffuse ground-glass opacity with interlobular and intralobular septal thickening creating a “crazy paving” pattern (black arrow). This is a nonspecific finding that can be seen in PERDS, DAH, viral pneumonia, and pulmonary edema. At the time of this scan, there was no lower lobe volume loss (white arrow). Despite steroid therapy, the patient’s symptoms worsened. (**b**) Axial image from a chest CT four days later shows increased lower lobe ground-glass opacity and septal thickening. The left major fissure is posteriorly displaced (white arrow) due to increasing lower lobe volume loss. (**c**) Axial image from a chest CT 11 days later shows findings of acute respiratory distress syndrome (ARDS) with extensive lower lobe predominant consolidation and ground-glass opacity with bronchial dilation (black arrow) and increasing lower lobe volume loss (white arrow). The patient developed pneumomediastinum (asterisk) due to barotrauma from intubation. (**d**) Diffuse alveolar hemorrhage (DAH) in a 32-year-old man 14 days status post-autologous HSCT. Axial CT image shows lower lobe predominant ground-glass opacity with a small focus of consolidation at the left base. Diffuse interlobular and intralobular septal thickening creating a “crazy paving” pattern (black arrow) is present. While this pattern has a broad differential diagnosis, DAH was confirmed on bronchoscopy.

**Figure 3 jcm-10-03227-f003:**
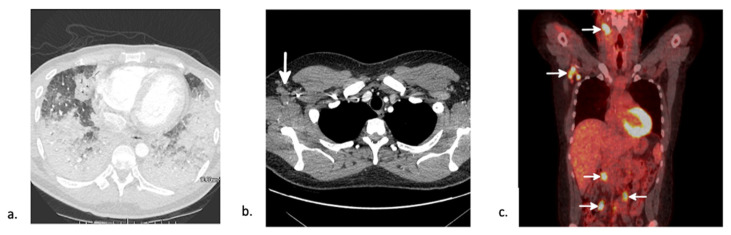
CT findings in early-post transplant non-infectious complications. (**a**) Axial CT in a 25-year-old man 89-days post-autologous HSCT for aplastic anemia shows lower lobe predominant consolidation. After an extensive work-up, the patient was diagnosed with idiopathic pneumonia syndrome (IPS). He died 24 days later. (**b**) Axial chest CT shows numerous enlarged right axillary lymph nodes (white arrow). (**c**) Coronal image from a subsequent positron emission tomography (PET)-CT shows increased FDG uptake in the right axillary nodes as well as uptake in right cervical and mesenteric lymph nodes (white arrows) consistent with a diagnosis of PTLD.

**Figure 4 jcm-10-03227-f004:**
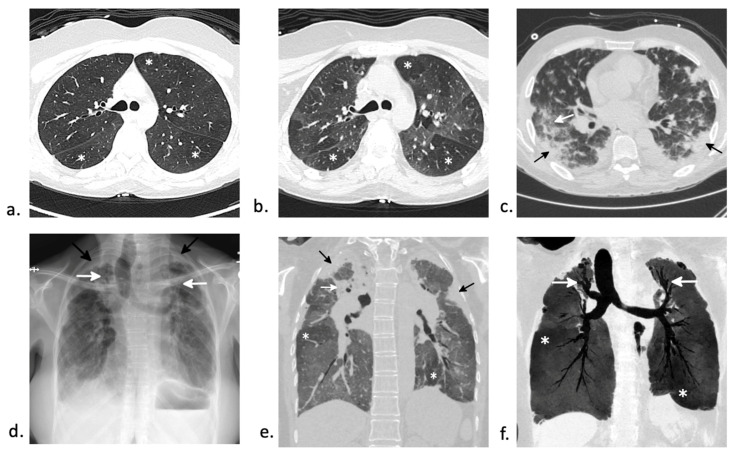
CT findings in late non-infectious complications. (**a**,**b**) Bronchiolitis obliterans syndrome (BOS) in a 42-year old woman two years after autologous HSCT. (a) Axial imaging during inspiration demonstrates a subtle mosaic attenuation with multiple areas of increased lucency (asterisks) adjacent to more normal lung. (**b**) On expiratory scan, the normal lung increases in attenuation. However, the areas with BO do not change in attenuation (asterisks) on expiration due to air trapping. (**c**) Chronic graft-versus-host disease (cGVHD) manifesting as organizing pneumonia (OP) in a 53-year-old man 5 months after autologous HSCT. Axial CT shows lower lobe and subpleural predominant consolidation and ground-glass opacity (black arrows) with a few scattered nodules. An “atoll” sign is present in the right lower lobe (white arrow). Extensive infectious work-up was negative and open lung biopsy showed organizing pneumonia. (**d**–**f**) Pleuroparenchymal fibroelastosis (PPFE) and bronchiolitis obliterans syndrome (BOS) in a 36-year-old woman 21 years after autologous HSCT for myelodysplastic syndrome. (**d**) Posteroanterior (PA) radiograph shows pronounced pleural thickening superiorly (black arrows) with superior displacement of the hila and upper lobe bronchiectasis (white arrows). (**e**) Corresponding coronal image from a chest CT better shows the exuberant superior pleural thickening extending inferior along the periphery of the upper lobes (black arrows). There is extensive upper lobe volume loss with superior displacement of the hila and areas of conglomerate perihilar and upper lobe fibrosis (white arrow). A mosaic attenuation is present in the lower lobes with areas of relative increased attenuation (asterisks). (**f**) A 10 mm thick minimum intensity projection image (MinIP) better shows the upper lobe predominant bronchiectasis (white arrows) and the areas of relative hypoattenuation in the lower lobes (asterisks). The patient underwent lung transplant which confirmed both PPFE and BO.

**Figure 5 jcm-10-03227-f005:**
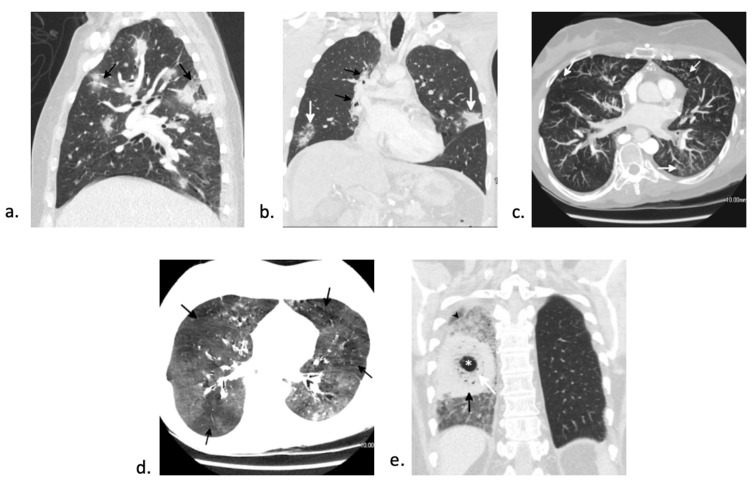
CT findings in early and late infectious complications. (**a**)Angioinvasive aspergillus in a 37-year-old woman status post-autologous HSCT for acute myeloid leukemia (AML). Sagittal chest CT shows numerous areas of nodular consolidation with surrounding ground-glass halos (black arrows) consistent with an angioinvasive infection. Aspergillus was confirmed on bronchoscopy. (**b**) Streptococcus pneumoniae infection in a 34-year-old man status post-autologous HSCT for Hodgkin lymphoma. Coronal chest CT shows nonspecific areas of consolidation in the left upper lobe and right lower lobe (white arrows). Streptococcus pneumoniae infection was confirmed on bronchoscopy. Paramediastinal fibrosis in the right lung (black arrows) due to radiation therapy for HL can be seen. (**c**,**d**) Nontuberculous mycobacterial infection (NTM) and bronchiolitis obliterans syndrome (BOS) in a 51-year-old woman three years after autologous stem cell transplant for acute lymphoblastic leukemia. (**c**) A 5 mm thick maximum intensity projection image (MIP) shows bronchial wall thickening and multifocal tree-in-bud nodularity (white arrows) due to NTM infection. (**d**) Corresponding 5 mm thick minimum intensity projection image (MinIP) at the same level nicely shows the pronounced mosaic attenuation (black arrows) due to BOS. (**e**) Mucor infection in a 49-year-old man nine months after autologous HSCT. Coronal image from a chest CT shows a mass-like lesion with dense circular consolidation (black arrow) with central necrosis manifesting as “bubbly”-appearing areas of ground-glass opacity (white arrow) and cavitation (asterisk). This “bird’s nest” sign is highly suggesting of mucormycosis. Surrounding ground-glass opacity with a “crazy paving” pattern (black arrowhead) is due to hemorrhage.

## Data Availability

No new data were created or analyzed in this study. Data sharing is not applicable to this article.

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
