# Peer review of "Pulmonary Complications in Hematopoietic Stem Cell Transplant Recipients—A Clinician Primer"

_jcm, 2021, doi:10.3390/jcm10153227_

Round 1
Reviewer 1 Report
This is a superb article - succinct and readable, and suitable for any non-HSCT specialist, including intensivists, internists and pulmonologists who doesn't regularly deal with HSCT patients. Images are outstanding - clear, and classic.
MINOR ISSUES:
Please define allogeneic, allo-HSCT and auto-HSCT for the non-expert.
Line 167 - Can PTLD be due to recipient EBV activation or new EBV infection, and/or as a result of other causes? Please elaborate.
Figure 1 should include a list of all the abbreviations used. Aeruginosa is mis-spelled.
Figure 3b - indicate adenopathy is due to PTLD.
Please avoid abbreviations used infrequently, and when they are used, ensure they're defined at first use.
Line 237 - contrast CT to evaluate angio-invasion may be more useful for detection of Aspergillus pneumonia (see Henzler, Sci Rep 2017 for one of many references).
Please briefly describe clinical and radiologic presentation of CMV pneumonia. You may want to indicate that HSV pneumonitis is frequently associated with DAD and pulmonary hemorrhage. Description of COVID-19 in this population is helpful.
What's the #118 at the end of the references? There are a few typos around [ parentheses around reference numbers, in the body text. Please check and correct.
Author Response
Please see the attachment
Reviewer 1:
- Please define allogeneic, allo-HSCT and auto-HSCT for the non-expert.
- Thank you for this feedback. In order to provide more background, we have added a paragraph on the comprehensive review of HSCT including definitions of allogeneic/allo-HSCT and auto-HSCT and a description of engraftment. This can be found in Section 2 of the revised paper titled Hematopoietic Stem Cell Transplant Overview starting line 34 and ending line 69 of the revised draft.
- Line 167 - Can PTLD be due to recipient EBV activation or new EBV infection, and/or as a result of other causes? Please elaborate.
- PTLD was further described as a heterogenous group of lymphoid disorders with clarification between PTLD in solid organs vs. HSCT. We have elaborated on reactivation vs new EBV infection, donor derived infection, as well as EBV-negative PTLD, which have all been cited in the recent literature. This was predominantly highlighted in lines 276-284.
- Figure 1 should include a list of all the abbreviations used. Aeruginosa is mis-spelled.
- Figure 1 was edited with the correct spelling of P aeruginosa. All abbreviations were included in the figure legend.
- Figure 3b - indicate adenopathy is due to PTLD.
- Indicated in the figure legend.
- Please avoid abbreviations used infrequently, and when they are used, ensure they're defined at first use.
- Abbreviations were reviewed and defined at first use throughout the paper.
- Line 237 - contrast CT to evaluate angio-invasion may be more useful for detection of Aspergillus pneumonia (see Henzler, Sci Rep 2017 for one of many references).
- Thank you for this insightful comment! The fungal pneumonia section has been addended to clarify between non-contrast CT and contrast CT findings, as well as the sensitivity and specificity for angioinvasive disease. This is noted in the revision lines 635-657.
- Please briefly describe clinical and radiologic presentation of CMV pneumonia.
- The CMV pneumonia section was revised to include presentation, diagnostic criteria, as well as imaging findings. These changes are noted from lines 734-744.
- You may want to indicate that HSV pneumonitis is frequently associated with DAD and pulmonary hemorrhage.
- Hemoptysis secondary to airway inflammation and tracheitis was described. Additionally, symptoms of HSV pneumonia were added in lines 754-757.
- Description of COVID-19 in this population is helpful.
- Description of COVID-19 symptoms were clarified with situations indicating mild-moderate disease in this patient population. Symptoms, laboratory findings and radiographic manifestations were added in lines 834-839.
- What's the #118 at the end of the references? There are a few typos around [ parentheses around reference numbers, in the body text. Please check and correct.
- This was a formatting issue. On edit, this was removed, and typos were corrected.
Reviewer 2 Report
The submitted article deals with a topic of current importance. This narrative review clearly explains the various lung pathological entities following a hematopoietic cell transplant and the previous papers are cited correctly.
My questions/suggestions to authors, before accepting the paper, are:
- here and there in the various paragraphs "open lung bipsy" is mentioned. Are you sure that a biopsy in thoracotomy or other open surgical approach is recommended in this kind of critical patients? Is video-assisted thoracoscopic surgery feasable? Are there any studies or reports that mention VATS? I think it must be clearly written in the manuscript, whether it is yes or no.
- likewise, a "Histopathologic confirmation" is mentioned in various paragraphs (e.g. page 6 line 227 - page 7 line 254 - page 7 line 269) but no technique was described (core needle biopsy? wedge resection of lung parenchyma? other?). In my opinion, a short paragraph summarizing all diagnostic techniques which lead to both a cytological and histological diagnosis should be added to the manuscript.
- Paragraph 7 ("Role of bronchoscopy in diagnosis") explains the use of endoscopy in diagnosis but it makes no mention of biopsies performed with EBUS (endobronchial ultrasound) bronchoscopy. Are there any studies that mention it? A good way to deal with this point could be adding a "diagnostic tool" paragraph that summarizes all techniques available, like said before.
Author Response
Please see the attachment
Reviewer 2:
- here and there in the various paragraphs "open lung biopsy" is mentioned. Are you sure that a biopsy in thoracotomy or other open surgical approach is recommended in this kind of critical patients? Is video-assisted thoracoscopic surgery feasible? Are there any studies or reports that mention VATS? I think it must be clearly written in the manuscript, whether it is yes or no.
- Thank you for this excellent feedback. You are absolutely correct in the notion that obtaining lung tissue has moved away from invasive surgical lung biopsy to less invasive methods including cryobiopsy, transbronchial biopsy and EBUS-guided transbronchial biopsy. As mentioned below, the role of bronchoscopy in the Diagnosis section has been expanded to include recent data regarding other methods of diagnosis, including surgical lung biopsies (both open thoracotomy and VATS), in lines 1028-1038.
- likewise, a "Histopathologic confirmation" is mentioned in various paragraphs (e.g. page 6 line 227 - page 7 line 254 - page 7 line 269) but no technique was described (core needle biopsy? wedge resection of lung parenchyma? other?). In my opinion, a short paragraph summarizing all diagnostic techniques which lead to both a cytological and histological diagnosis should be added to the manuscript.
- The authors of this paper agree with this sentiment and expanded significantly on the ending paragraph of this paper to include diagnostic methods of obtaining tissue samples. This section will in totality be between lines 983-1038 in the new revision.
- Paragraph 7 ("Role of bronchoscopy in diagnosis") explains the use of endoscopy in diagnosis but it makes no mention of biopsies performed with EBUS (endobronchial ultrasound) bronchoscopy. Are there any studies that mention it? A good way to deal with this point could be adding a "diagnostic tool" paragraph that summarizes all techniques available, like said before.
- The role of bronchoscopy paragraph was expanded into a section on diagnostic tools. More data regarding EBUS in this patient population was sought out. Although there appears to be excellent data in its diagnosis of GGOs in the general patient population, there is less evidence in those who are immunosuppressed. There is a recent publication by Buoso et al. that described its diagnostic yield for infectious lesions in immunocompromised pediatric patients. This is described in lines 1018-1027 of the new draft. Of note, the literature on EBUS in our specific HSCT population for post-HSCT pulmonary complications is limited.
Round 2
Reviewer 2 Report
Thanks to the changes made to the paper, I believe it is now worthy of publication in its current form.
I congratulate the authors for their work.